# GRAPH CONTRASTIVE LEARNING FOR SKELETON-BASED ACTION RECOGNITION

**Xiaohu Huang** [1,2]* **Hao Zhou** [2]† **Jian Wang** [2] **Haocheng Feng** [2] **Junyu Han** [2]
**Errui Ding** [2] **Jingdong Wang** [2] **Xinggang Wang** [1] **Wenyu Liu** [1] **Bin Feng** [1]†

[1] School of EIC, Huazhong University of Science & Technology
[2] Department of Computer Vision Technology (VIS), Baidu Inc., China
{huangxiaohu,xgwang,liuwy,fengbin}@hust.edu.cn
{zhouhao14,wangjian33,fenghaocheng,hanjunyu}@baidu.com
{dingerrui,wangjingdong}@baidu.com

## ABSTRACT

In the field of skeleton-based action recognition, current top-performing graph convolutional networks (GCNs) exploit intra-sequence context to construct adaptive graphs for feature aggregation. However, we argue that such context is still *local* since the rich cross-sequence relations have not been explicitly investigated. In this paper, we propose a graph contrastive learning framework for skeleton-based action recognition (*SkeletonGCL*) to explore the *global* context across all sequences. In specific, SkeletonGCL associates graph learning across sequences by enforcing graphs to be class-discriminative, *i.e.,* intra-class compact and inter-class dispersed, which improves the GCN capacity to distinguish various action patterns. Besides, two memory banks are designed to enrich cross-sequence context from two complementary levels, *i.e.,* instance and semantic levels, enabling graph contrastive learning in multiple context scales. Consequently, SkeletonGCL establishes a new training paradigm, and it can be seamlessly incorporated into current GCNs. Without loss of generality, we combine SkeletonGCL with three GCNs (2S-ACGN, CTR-GCN, and InfoGCN), and achieve consistent improvements on NTU60, NTU120, and NW-UCLA benchmarks. The source code will be available at https://github.com/OliverHxh/SkeletonGCL.

## 1 INTRODUCTION

Graph convolutional networks (GCNs) have been widely applied in skeleton-based action recognition since they can naturally process non-grid skeleton sequences. For GCN-based methods, how to effectively learn the graphs remains a core and challenging problem. In particular, ST-GCN (Yan et al., 2018) is a milestone work, using pre-defined graphs to extract the action patterns. However, the pre-defined graphs only enable each joint to access the fixed neighboring joints but are hard to capture long-range dependency adaptively. Therefore, a mainstream of subsequent works (Li et al., 2019; Shi et al., 2019; Zhang et al., 2020b;a; Ye et al., 2020; Chen et al., 2021b; Chi et al., 2022) take efforts to solve this issue by generating adaptive graphs. The adaptive graphs can dynamically aggregate the features within each sequence and thus show significant advantages in performance comparison.

Generally, adaptive graphs are constructed by using intra-sequence context. However, such context will still be "local" when viewing the cross-sequence information as an available context. Therefore, we wonder: *Is it possible to involve the cross-sequence context in graph learning?* To find out the answer, in Fig. 1, we visualize the adaptive graphs learned from sequences of two easily confused classes ("point to something" and "take a selfie"). The graphs are learned by a strong GCN, *i.e.,*

---

*Work done when Xiaohu Huang was an intern at Baidu VIS.
†Corresponding authors.

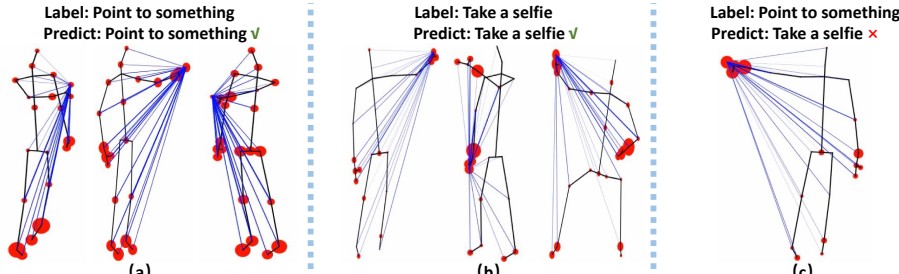

Figure 1: **Graph visualization of sequences from two easily confused classes** ("point to something" and "take a selfie"). The graphs are learned by CTR-GCN (Chen et al., 2021b). We take the tip of the hand that does the action as the anchor. The size of the red circles and the width of the blue lines both denote the strengths of connections between joints. For simplicity, only representative frames are visualized. (a) Three sequences from class "point to something" are correctly classified, where the graphs contain connections to the body joints. (b) Three sequences from class "take a selfie" are correctly classified, where the graphs highly emphasize the connections to the hands, while the connections to the body are suppressed. (c) A sequence from class "point to something" is misclassified as "take a selfie", whose graph resembles the graphs in (b), but is dissimilar from graphs in (a). Hence, we realize that the class-ambiguous graph representations would make negative impacts on recognition performance.

CTR-GCN (Chen et al., 2021b). From the visualization, we find that (1) For sequences that are correctly classified in Fig. 1 (a) and Fig. 1 (b), the learned graphs in the same class look similar, while graphs in different classes have distinct differences. (2) For a misclassified sequence in Fig. 1 (c), the learned graph resembles the graphs from the misclassified class more than those from the ground truth class. These observations remind us that graph learning in current adaptive GCNs can implicitly learn class-specific graph representations to some extent. But without explicit constraints, it leads to class-ambiguous representations in some cases, thereby affecting the GCN capacity to discriminate classes (in Tab. 9 of Sec. 4.4, we provide quantitative results to further support our hypothesis). Therefore, we speculate that if the cross-sequence semantic relations are incorporated as guidance in graph learning, the class-ambiguity issue will be alleviated and the graph representations will better express individual characteristics of actions.

In recent years, contrastive learning has achieved great success in self-supervised representation learning (He et al., 2020; Chen et al., 2020; 2021a). It studies cross-sample relations in the dataset. The essence of contrastive learning is "comparing", which pulls together the feature embedding from positive pairs and pushes away the feature embedding from negative pairs.

Based on the analysis above and the advances in contrastive learning, we propose a graph contrastive learning framework for skeleton-based action recognition in the fully-supervised setting, dubbed **SkeletonGCL**. Instead of just using the local information within each sequence, SkeletonGCL explores the cross-sequence global context to improve graph learning. The core idea is to pull together the learned graphs from the same class while pushing away the learned graphs from different classes. Since graphs can reveal the action patterns of sequences, enforcing graph consistency in the same class and inconsistency among different classes helps the model understand various motion modes. In addition, to enrich the cross-sequence context, we build memory banks to store the graphs from historical sequences. In specific, an instance-level memory bank stores the sequence-wise graphs, which hold the individual properties of each sequence. In contrast, a semantic-level memory bank stores the aggregation of graphs from each class, which contains the class-level representation. The two banks are complementary to each other, enabling us to leverage more samples. SkeletonGCL can be seamlessly combined with existing GCNs. Eventually, we combine SkeletonGCL with three GCNs (2S-AGCN (Shi et al., 2019), CTR-GCN (Chen et al., 2021b), and InfoGCN (Chi et al., 2022)), and conduct experiments on three popular datasets (NTU60 (Shahroudy et al., 2016), NTU120 (Liu et al., 2019) and NW-UCLA (Wang et al., 2014)). SkeletonGCL achieves consistent improvements with these models using different testing protocols (single-modal or multi-modal) on all three datasets, which widely demonstrates the effectiveness of our design. Notably, SkeletonGCL only introduces a small amount of training consumption but has no impact at the test stage.

Though there exist some works that apply contrastive learning in skeleton-based action recognition (Li et al., 2021; Guo et al., 2022; Mao et al., 2022), our method differs from them as follows: (1) The previous methods took pooled feature vectors to conduct contrastive learning as in (He et al., 2020; Chen et al., 2020), where the structural properties in skeletons are lost. In contrast, SkeletonGCL uses graphs to contrast, which maintains the structure details of skeletons and offers high-order connection information between joints. (2) The previous methods used memory banks to store instance-level representations only. Differently, our memory banks store both instance-level and semantic-level representations, allowing us to leverage context from individual sequences and class-specific aggregations, which are complementary to each other. (3) The previous methods were used in the pre-training stage, while SkeletonGCL is incorporated into the fully-supervised setting without extra pre-training cost.

Summarily, the contribution of this paper can be concluded as follows:

- We present a new perspective for graph learning of GCN models in skeleton-based action recognition. In specific, we propose to make use of the cross-sequence context to guide graph learning, whose goal is to enforce graphs to be intra-class compact and inter-class dispersed.
- Motivated by the advances in contrastive learning, we smoothly combine the ideas of contrastive learning and cross-sequence graph learning together, then propose a new training paradigm for skeleton-based action recognition, called SkletonGCL. SkeletonGCL incorporates an instance-level and a semantic-level memory bank to enrich the cross-sequence context comprehensively. Besides, it can be seamlessly incorporated into current GCNs.
- SkeletonGCL achieves consistent improvements combined with three GCNs (2S-AGCN, CTR-GCN, and InfoGCN) on three popular benchmarks (NTU60, NTU120, NW-UCLA) using both single-modal and multi-modal testing protocols. In addition, SkeletonGCL is training-efficient and has no impact at the test stage.

## 2 RELATED WORKS

### 2.1 SKELETON-BASED ACTION RECOGNITION

Skeleton-based action recognition is to classify actions from sequences of estimated key points. The early deep-learning methods applied convolution neural networks (CNNs) (Chéron et al., 2015; Liu et al., 2017b) or recurrent neural networks (RNNs) (Du et al., 2015; Lev et al., 2016; Wang & Wang, 2017; Liu et al., 2017a) to model the skeletons, but they could not explicitly explore the topological structure of skeletons, thus the performances were limited. Recently, PoseC3D (Duan et al., 2022) revisited the CNN-based method by stacking the heatmaps as 3D volumes, which maintained the spatial-temporal properties of skeletons and obtained marginal performance improvements. In the past few years, the mainstream works in skeleton-based action recognition were GCN models. ST-GCN (Yan et al., 2018) was the first work that adopted GCN as the feature extractor, which heuristically designed fixed graphs to model the skeletons. The follow-up methods proposed spatial-temporal graphs (Liu et al., 2020), multi-scale graph convolutions (Chen et al., 2021c), channel-decoupled graphs (Chen et al., 2021b; Cheng et al., 2020a) and adaptive graphs (Li et al., 2019; Shi et al., 2019; Ye et al., 2020; Zhang et al., 2020b; Chen et al., 2021b; Chi et al., 2022) to improve the capacity of GCNs. Tracking the development of GCN-based methods, we find that graph learning has always been a core problem and now the adaptive GCNs are leading since they can model the intrinsic topology of skeletons.

However, current adaptive GCNs generated the graphs based on the local context within each sequence, where the cross-sequence relations have been neglected. In contrast, we propose to explore the cross-sequence global context to shape graph representations. In this way, the learned graphs can not only describe the individual characteristics within each sequence but also emphasize the similarity and dissimilarity of motion patterns across sequences.

### 2.2 CONTRASTIVE LEARNING

In recent years, numerous representation learning methods (Wu et al., 2018; Oord et al., 2018; He et al., 2020; Chen et al., 2020; Wang et al., 2021) with contrastive learning have emerged, especially

in self-supervised representation learning. The key idea is to pull together the positive pairs and push away the negative pairs in the feature space. Generally, the features are vectors obtained from feature extractors followed by a pooling layer. A standard approach to obtaining the positive pairs is augmenting an original sample into two different views. The negative samples are selected randomly or using hard mining strategies (Khosla et al., 2020; Robinson et al., 2020; Kalantidis et al., 2020). To increase the capacity of negative samples, the memory bank mechanism was devised in (He et al., 2020; Misra & Maaten, 2020) to store more negative instances. By contrasting positive pairs against negative pairs, the model can learn to focus on semantic representations.

In the field of skeleton-based action recognition, prior works (Li et al., 2021; Mao et al., 2022; Guo et al., 2022) proposed to apply contrastive learning in the pre-training stage by roughly following the frameworks mentioned above. CrossCLR (Li et al., 2021) mined positive pairs in the data space and explored the cross-modal distribution relationships. Further, CMD (Mao et al., 2022) transferred the cross-modal knowledge in a distillation manner. And AimCLR (Guo et al., 2022) used extreme augmentations to improve the representation universality.

Compared with the above methods, we use graph representations to contrast instead of using pooled feature vectors. Meanwhile, we establish two different memory banks at complementary levels, *i.e.,* instance and semantic levels, to enrich the context scales. Besides, the proposed method is used with GCNs under the fully-supervised setting, which requires no pre-training procedure.

## 3 METHOD

### 3.1 PRELIMINARY

We denote a human skeleton as a vertex set $\mathcal{V} = \{v_1, v_2, ..., v_N\}$, where $N$ denotes the number of vertices. For each vertex $v_i$, the feature dimension is set as $C$. Hence, a skeleton sequence with $T$ frames can be denoted as $\mathbf{X} \in \mathbb{R}^{T \times N \times C}$. Graph topology is used to represent the correlations between joints, formulated as $\mathbf{g}$.

**GCNs in Skeleton-Based Action Recognition.** Generally, GCN models alternatively apply graph convolutions and temporal convolutions to extract the spatial configuration and motion pattern of skeletons, respectively. The graph $\mathbf{g}$ is vital for graph convolutions since it determines the message passing among joints. In current adaptive GCNs, $\mathbf{g}$ is learned within each sequence and has different sizes, *e.g.,* $\mathbf{g} \in \mathbb{R}^{K_S \times N \times N}$ in 2S-AGCN (Shi et al., 2019) and $\mathbf{g} \in \mathbb{R}^{K_S \times C \times N \times N}$ in CTR-GCN (Chen et al., 2021b). The $K_S$ denotes the number of sub-graphs, normally set as 3. In general, the graph convolution is defined as:

$$\mathbf{X_S} = \sum_{k=1}^{K_S} \mathbf{g}^k \mathbf{X} \mathbf{W_S^k}, \tag{1}$$

where $\mathbf{X_S} \in \mathbb{R}^{T \times N \times C'}$ denotes the spatial extracted feature with $C'$ channels, and $\mathbf{W_S} \in \mathbb{R}^{K_S \times C \times C'}$ denotes the spatial feature transformation filters. Next, temporal convolutions are applied on $\mathbf{X_S}$, producing motion extracted feature $\mathbf{X_T} \in \mathbb{R}^{T \times N \times C'}$. After stacking layers of graph convolutions and temporal convolutions, a global average pooling (GAP) layer summarizes the global features, then a classification head (fully-connected layer) followed by a Softmax activation function is applied to obtain the class prediction $\hat{\mathbf{y}} \in \mathbb{R}^{C_k}$, where $C_k$ denotes the number of classes. Finally, a cross-entropy loss $\mathcal{L}_{\text{CE}}$ supervises the class prediction with the ground truth label $\mathbf{y}$ as follows:

$$\mathcal{L}_{\text{CE}} = -\sum_i \mathbf{y_i} \log \hat{\mathbf{y}_i} \tag{2}$$

**Self-Supervised Contrastive Learning.** In the context of self-supervised contrastive learning, each input sample is processed by data augmentations to produce a positive pair: $I$ and $I^+$. Through a feature extraction network, $I$ and $I^+$ are transformed into feature vectors $\mathbf{f} \in \mathbb{R}^D$ and $\mathbf{f}^+ \in \mathbb{R}^D$. As for the negative samples, they are selected from the dataset excluding $I$ and $I^+$, represented as a negative set $\mathcal{N}^-$. Each negative in $\mathcal{N}^-$ is denoted as $\mathbf{f}^- \in \mathbb{R}^D$. The similarity between two feature vectors is calculated as $sim(\mathbf{f}^+, \mathbf{f}^-) = \frac{\mathbf{f}^+ \mathbf{f}^-}{\|\mathbf{f}^+\| \|\mathbf{f}^-\|}$. InfoNCE (Gutmann & Hyvärinen, 2010; Oord

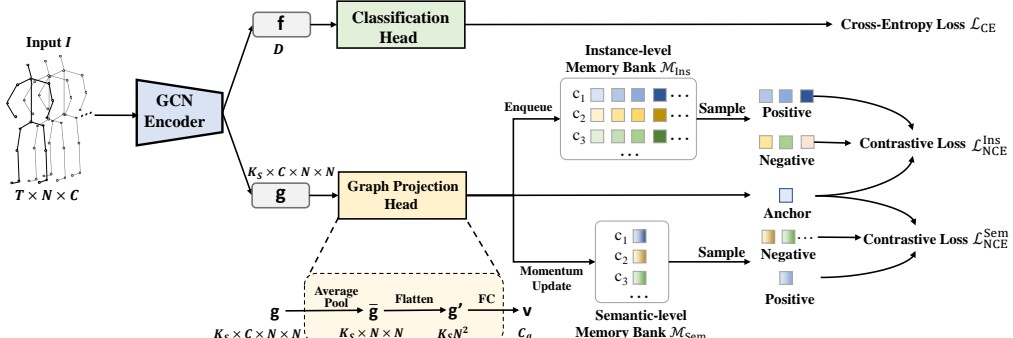

Figure 2: **Overview of SkeletonGCL.** An input skeleton sequence $I$ is fed into a GCN encoder, producing a feature vector $\mathbf{f}$ for classification and a learned graph $\mathbf{g}$ for graph contrastive learning. The graph $\mathbf{g}$ is embedded into a vector by a projection head. And two memory banks are built to store the embedded graphs. From the memory banks, we sample the positives and negatives according to the labels, then perform contrastive loss. The memory banks are only used in the training stage but will be removed during the testing stage.

et al., 2018) is widely adopted for contrastive learning, which is formulated as:

$$\mathcal{L}_{\text{NCE}} = -\log \frac{sim(\mathbf{f}, \mathbf{f}^+)/\tau}{sim(\mathbf{f}, \mathbf{f}^+)/\tau + \sum_{\mathbf{f}^- \in \mathcal{N}^-} sim(\mathbf{f}, \mathbf{f}^-)/\tau}, \tag{3}$$

where temperature $\tau > 0$ is a hyper-parameter.

## 3.2 Graph Contrastive Learning

The proposed SkeletonGCL is illustrated in Fig. 2. The framework consists of two branches, where the classification branch is parallel to the graph contrast branch. Taking a skeleton sequence $I$ as input, the GCN encoder outputs a feature vector $\mathbf{f}$ for classification and a graph $\mathbf{g}$ for graph contrast.

**Graph Projection Head.** In order to contrast the graphs in a common feature space, we embed the graphs into vectors by a graph projection head. The projection heads for different GCNs are similar (see App. 6.1 for details). In Fig. 2, taking the graph $\mathbf{g} \in \mathbb{R}^{K_S \times C \times N \times N}$ learned in CTR-GCN (Chen et al., 2021b) as an example, we first squeeze $\mathbf{g}$ along the channel dimension by an average pooling layer into $\overline{\mathbf{g}} \in \mathbb{R}^{K_S \times N \times N}$. Then, we flatten graph $\overline{\mathbf{g}}$ into a 1D vector as $\mathbf{g}' \in \mathbb{R}^{K_S N^2}$ and project $\mathbf{g}'$ into a vector $\mathbf{v} \in \mathbb{R}^{C_g}$ by an FC layer $\mathbf{W_G} \in \mathbb{R}^{K_S N^2 \times C_g}$. Since different channels in $\mathbf{W_G}$ are specific to different vertices in the graph, the graph projection is vertex-aware and thus can encode the structures of skeletons. Afterward, we update two memory banks with $\mathbf{v}$. The memory banks are illustrated in Fig. 2, and detailed next.

**Memory Bank.** To enrich the cross-sequence context, we build memory banks to store the cross-batch graphs. In specific, two memory banks are constructed, *i.e.,* an instance-level memory bank $\mathcal{M}_{\text{Ins}} \in \mathbb{R}^{C_k \times P \times C_g}$ and a semantic-level memory bank $\mathcal{M}_{\text{Sem}} \in \mathbb{R}^{C_k \times C_g}$. $P$ denotes the number of instances stored for each class in $\mathcal{M}_{\text{Ins}}$. Particularly, each element in $\mathcal{M}_{\text{Ins}}$ denotes a graph instance from a class. In contrast, each element in $\mathcal{M}_{\text{Sem}}$ denotes the graph aggregation of a class. Therefore, the two memory banks are on complementary levels, where the instance-level memory bank emphasizes the instance discrimination of each sequence, while the semantic-level memory bank covers the class properties across sequences.

We update $\mathcal{M}_{\text{Ins}}$ in a first-in-first-out manner, which maintains the number of instances for each class as $P$. As for $\mathcal{M}_{\text{Sem}}$, we use a momentum update strategy, which integrates the graphs of the same class from the current timestamp and all previous timestamps, regarded as a long-term representation. The momentum update is defined as follows:

$$\mathbf{m}_{\text{sem}}^{c_*} \leftarrow \alpha \mathbf{m}_{\text{sem}}^{c_*} + (1-\alpha)\mathbf{v}, \tag{4}$$

where $\mathbf{m}_{\text{sem}}^{c_*}$ is the representation for class $c_*$, $c_*$ is the class label for the input $I$ and $0 < \alpha < 1$ is a hyper-parameter.

**Loss.** To achieve the graph contrast, we sample positives and negatives from the memory banks $\mathcal{M}_{\text{Ins}}$ and $\mathcal{M}_{\text{Sem}}$. For $\mathcal{M}_{\text{Ins}}$, vector $\mathbf{v}$ is set as the anchor, hence samples in the positive set $\mathcal{N}_{\text{Ins}}^{+}$ are with label $c_*$, and samples in the negative set $\mathcal{N}_{\text{Ins}}^{-}$ are with different labels. Consequently, the InfoNCE loss in Eq. 3 can be rewritten as:

$$\mathcal{L}_{\text{NCE}}^{\text{Ins}} = -\sum_{\mathbf{v}^+ \in \mathcal{N}_{\text{Ins}}^+} \log \frac{sim(\mathbf{v}, \mathbf{v}^+)/\tau}{sim(\mathbf{v}, \mathbf{v}^+)/\tau + \sum_{\mathbf{v}^- \in \mathcal{N}_{\text{Ins}}^-} sim(\mathbf{v}, \mathbf{v}^-)/\tau}, \tag{5}$$

$$\mathcal{L}_{\text{NCE}}^{\text{Sem}} = -\sum_{\mathbf{v}^+ \in \mathcal{N}_{\text{Sem}}^+} \log \frac{sim(\mathbf{v}, \mathbf{v}^+)/\tau}{sim(\mathbf{v}, \mathbf{v}^+)/\tau + \sum_{\mathbf{v}^- \in \mathcal{N}_{\text{Sem}}^-} sim(\mathbf{v}, \mathbf{v}^-)/\tau}. \tag{6}$$

$\mathcal{L}_{\text{NCE}}^{\text{Ins}}$ leverages multiple positives compared with Eq. 3 by using label information, which mines more semantic-related samples. Similarly, we can define the InfoNCE loss $\mathcal{L}_{\text{NCE}}^{\text{Sem}}$, which is specific for the memory bank $\mathcal{M}_{\text{Sem}}$. Summarily, the overall contrastive loss is written as follows:

$$\mathcal{L}_{\text{NCE}} = \mathcal{L}_{\text{NCE}}^{\text{Ins}} + \mathcal{L}_{\text{NCE}}^{\text{Sem}}. \tag{7}$$

And the overall loss function is defined as follows:

$$\mathcal{L} = \mathcal{L}_{\text{NCE}} + \mathcal{L}_{\text{CE}}. \tag{8}$$

**Hard Sampling.** As the training continues, most samples become too easy, which contribute less to the training. Therefore, methods in (Tabassum et al., 2022; Robinson et al., 2020; Kalantidis et al., 2020; Wang et al., 2021) are proposed to use hard mining strategies to focus on informative samples. In this paper, considering the massive number of instances in $\mathcal{M}_{\text{Ins}}$, contrasting with all these instances naturally leads to redundancy and hinders the training. To alleviate this issue, we propose to mine hard examples in $\mathcal{M}_{\text{Ins}}$. Specifically, we take the similarity calculation $sim(\mathbf{v}, \mathbf{v}')$ as a criterion to evaluate hardness. Harder positives are with lower similarities, and harder negatives are with higher similarities. In total, for $\mathcal{M}_{\text{Ins}}$, we select $K_{\text{H}}^{+}$ hardest positive examples, $K_{\text{H}}^{-}$ hardest negative examples, and $K_{\text{R}}^{-}$ random negative examples.

## 4 EXPERIMENTS

### 4.1 DATASETS

**NTU RGB+D**. NTU RGB+D (NTU60) (Shahroudy et al., 2016) is a large-scale skeleton-based action recognition dataset, which contains 60 action classes and 56,880 sequences. Each sequence is annotated as skeletons with 25 joints. All the sequences are performed by 40 subjects and filmed by 3 Kinect cameras from three different views. Generally, two protocols are used to evaluate the performances: (1) cross-subject (X-Sub): train data are performed by 20 subjects, and test data are performed by other 20 subjects. (2) cross-view (X-View): train data from cameras 2 and 3, and test data from camera 1.

**NTU RGB+D 120.** NTU RGB+D 120 (NTU120) (Liu et al., 2019) is an extension of NTU RGB+D dataset, which newly includes 57,367 skeletons of 60 extra classes. All the sequences are performed by 106 subjects and filmed by three cameras from three different views. In addition, NTU RGB+D 120 has 32 setups, where each denotes a unique location. Generally, two protocols are used to evaluate the performances: (1) cross-subject (X-Sub): train data are performed by 53 subjects, and test data are performed by other 53 subjects. (2) cross-setup (X-Set): train data are samples with even setup IDs, and test data are samples with odd setup IDs.

**Northwestern-UCLA**. Northwestern-UCLA (NW-UCLA) dataset (Wang et al., 2014) contains 1494 sequences from 10 action classes. Each sequence is annotated as skeletons with 20 joints. All sequences are performed by 10 subjects and filmed by three Kinect cameras from different views. We follow the official evaluation protocol: train data are captured by the first two cameras, and test data are captured by the third camera.

### 4.2 IMPLEMENTATION DETAILS

To thoroughly validate SkeletonGCL, we take three GCNs (2S-AGCN (Shi et al., 2019), CTR-GCN (Chen et al., 2021b), and InfoGCN (Chi et al., 2022)) as baseline models. For CTR-GCN and

Table 1: Top-1 accuracy comparison (%) with the state-of-the-art methods on NTU 60 and NTU120 datasets. The numbers in gray indicate the results reported in their papers. * indicates that we retrain the models using their officially released code. Particularly, 2S-AGCN is retrained using a stronger train recipe from CTR-GCN.

| Dataset | NTU 60 | | | | | | | | NTU 120 | | | | | | | |
|---|---|---|---|---|---|---|---|---|---|---|---|---|---|---|---|---|
| Setting | X-Sub | | | | X-View | | | | X-Sub | | | | X-Set | | | |
| Method/Modality | $J$ | $B$ | $J+B$ | $4S$ | $J$ | $B$ | $J+B$ | $4S$ | $J$ | $B$ | $J+B$ | $4S$ | $J$ | $B$ | $J+B$ | $4S$ |
| SGCN (Zhang et al., 2020a) | - | - | 89.0 | - | - | - | 94.5 | - | - | - | 79.2 | - | - | - | 81.5 | - |
| ST-TR-GCN (Plizzari et al., 2021) | 89.2 | - | 90.3 | - | 95.8 | - | 96.3 | - | 82.7 | - | 85.1 | - | 85.0 | - | 87.1 | - |
| Shift-GCN (Cheng et al., 2020b) | 87.8 | - | 89.7 | 90.7 | 95.1 | - | 96.0 | 96.5 | 80.9 | - | 85.3 | 85.9 | 83.2 | - | 86.6 | 87.6 |
| DC-GCN+ADG (Cheng et al., 2020a) | - | - | 90.8 | - | - | - | 96.6 | - | - | - | 86.5 | - | - | - | 88.1 | - |
| Dynamic GCN (Ye et al., 2020) | - | - | - | 91.5 | - | - | - | 96.0 | - | - | - | 87.3 | - | - | - | 88.6 |
| MS-G3D (Liu et al., 2020) | 89.4 | 90.1 | 91.5 | - | 95.0 | 95.3 | 96.2 | - | - | - | 86.9 | - | - | - | 88.4 | - |
| MST-GCN (Chen et al., 2021c) | 89.0 | 89.5 | 91.1 | 91.5 | 95.1 | 95.2 | 96.4 | 96.6 | 82.8 | 84.8 | 87.0 | 87.5 | 84.5 | 86.3 | 88.3 | 88.8 |
| 2S-AGCN (Shi et al., 2019) | - | - | 88.5 | - | 93.7 | 93.2 | 95.1 | - | - | - | - | - | - | - | - | - |
| 2S-AGCN* (Shi et al., 2019) | 88.9 | 89.2 | 91.0 | 91.5 | 94.5 | 94.1 | 95.7 | 95.9 | 84.0 | 85.1 | 87.8 | 88.2 | 85.3 | 86.3 | 89.0 | 89.6 |
| 2S-AGCN* w/SkeletonGCL | 89.9 | 90.0 | 91.6 | 92.2 | 95.0 | 94.4 | 96.1 | 96.4 | 84.7 | 86.0 | 88.4 | 88.7 | 86.1 | 86.8 | 89.7 | 90.2 |
| CTR-GCN (Chen et al., 2021b) | - | - | - | 92.4 | - | - | - | 96.8 | - | 85.7 | 88.7 | 88.9 | - | 87.5 | 90.1 | 90.6 |
| CTR-GCN* (Chen et al., 2021b) | 89.8 | 90.2 | 92.0 | 92.4 | 94.8 | 94.8 | 96.3 | 96.8 | 84.9 | 85.7 | 88.7 | 88.9 | 86.7 | 87.5 | 90.1 | 90.5 |
| CTR-GCN* w/SkeletonGCL | 90.8 | 91.1 | 92.6 | 93.1 | 95.3 | 95.4 | 96.6 | 97.0 | 85.6 | 86.9 | 89.2 | 89.5 | 87.3 | 88.2 | 90.5 | 91.0 |
| InfoGCN (Chi et al., 2022) | 89.8 | 90.6 | 91.6 | 92.7 | 95.2 | 95.5 | 96.5 | 96.9 | 85.1 | 87.3 | 88.5 | 89.4 | 86.3 | 88.5 | 89.7 | 90.7 |
| InfoGCN* (Chi et al., 2022) | 89.4 | 90.6 | 91.3 | 92.3 | 95.2 | 95.4 | 96.2 | 96.7 | 84.2 | 86.9 | 88.2 | 89.2 | 86.3 | 88.5 | 89.4 | 90.7 |
| InfoGCN* w/SkeletonGCL | 90.1 | 91.0 | 91.9 | 92.8 | 95.5 | 95.7 | 96.6 | 97.1 | 85.2 | 87.4 | 88.8 | 89.8 | 87.2 | 88.7 | 90.0 | 91.2 |

InfoGCN, we follow their training recipes. Particularly, for 2S-AGCN, since its training recipe is out of date, we borrow the training recipe from CTR-GCN, which effectively improves its baseline performance. $P$, the number of stored instances for each class in $\mathcal{M}_{\text{Ins}}$, is set as 684 on NTU60 and NTU120, and 342 on NW-UCLA. The dimension of graph vector $C_g$ is set to 256. For all datasets, the number of sampling examples $K_H^+$, $K_H^-$, and $K_R^-$ are set as 128, 512, and 512, respectively. For different models used in different modalities, we experiment with temperature $\tau$ of 0.5, 0.8, 1.0, and 1.5, and choose the best one. The hyper-parameter $\alpha$ for momentum updating is set as 0.85. Besides, we fix the random seed to ensure experiment reproducibility. All experiments are conducted using a single NVIDIA V100 GPU.

## 4.3 COMPARED WITH THE STATE-OF-THE-ART

In this section, we combine our method with three GCNs, and compare them with the state-of-the-art (SoTA) methods. In Tab. 1 and Tab. 2, we list current SoTA methods in skeleton-based action recognition except PoseC3D (Duan et al., 2022). PoseC3D is a promising CNN-based method, but it uses non-official skeleton data and applies a multi-crop test protocol (GCN methods typically use one crop), which are unfair for comparison here. In evaluation, four modalities are used: "joint stream" (*J*) denotes the joint coordinates, "bone stream" (*B*) denotes the coordinate difference between spatially connected joints, "joint motion" (*J-M*) denotes the coordinate difference between temporally adjacent frames, and "bone motion" (*B-M*) denotes the bone difference between temporally adjacent frames. The 4-stream ensemble (*4S*) denotes using the four modalities together. Following the widely-adopted protocol, we evaluate models using $J$, $B$, $J + B$, and $4S$ modalities.

**NTU60 and NTU120.** Tab. 1 lists the results on NTU60 and NTU120. From the results, we find that: (1) Combined with SkeletonGCL, all three baseline models achieve solid improvements on these two benchmarks over different settings and modalities. Taking the *J* modality on NTU60 X-Sub as an example, 2S-AGCN improves by 1.0% (88.9% to 89.9%), CTR-GCN improves by 1.0% (89.8% to 90.8 %), and InfoGCN improves by 0.7% (89.4% to 90.1%). Considering NTU60 is an extensively-benchmarked dataset, such improvements are quite hard. (2) With SkeletonGCL, CTR-GCN and InfoGCN can set new SoTA performance.

**NW-UCLA.** Tab. 2 lists the results on NW-UCLA. SkeletonGCL can still achieve consistent improvements based on the three models. And new state-of-the-art performances are achieved by combining SkeletonGCL with CTR-GCN and InfoGCN.

Table 2: Top-1 accuracy Comparison (%) with the state-of-the-art methods on the NW-UCLA dataset. Numbers in gray denote the results reported in their papers. * indicates that we retrain the models using their officially released codes. Particularly, 2S-AGCN is retrained using a stronger train recipe from CTR-GCN.

| Dataset | NW-UCLA | | | |
|---|---|---|---|---|
| Method/Modality | *J* | *B* | *J+B* | *4S* |
| AGC-LSTM (Si et al., 2019) | 93.3 | - | - | - |
| DC-GCN+ADG (Cheng et al., 2020a) | - | - | 95.3 | - |
| Shift-GCN (Cheng et al., 2020b) | 92.5 | - | 94.2 | 94.6 |
| 2S-AGCN (Shi et al., 2019) | - | - | - | - |
| 2S-AGCN* (Shi et al., 2019) | 92.0 | 92.2 | 95.0 | 95.5 |
| 2S-AGCN* *w*/SkeletonGCL | **92.6** | **93.0** | **95.7** | **96.3** |
| CTR-GCN (Chen et al., 2021b) | - | - | - | 96.5 |
| CTR-GCN* (Chen et al., 2021b) | 94.6 | 91.8 | 94.2 | 96.5 |
| CTR-GCN* *w*/SkeletonGCL | **95.1** | **95.0** | **95.9** | **96.8** |
| InfoGCN (Chi et al., 2022) | 94.0 | 95.3 | 96.3 | 96.6 |
| InfoGCN* (Chi et al., 2022) | 93.8 | 94.2 | 95.5 | 96.1 |
| InfoGCN* *w*/SkeletonGCL | **94.8** | **94.6** | **96.1** | **96.8** |

## 4.4 DIAGNOSTIC EXPERIMENTS

In this section, we conduct diagnostic experiments to verify the design of SkeletonGCL. Otherwise stated, we use CTR-GCN as the GCN encoder to perform the experiments on the NTU60 dataset under the X-Sub setting using the joint modality (*J*). See App. 6.2 for more diagnostic experiments.

Table 3: Comparison of intra-batch and inter-batch contrast.

| Model (CTR-GCN) | Acc (%) |
|---|---|
| Baseline (*w/o* contrast) | 89.8 |
| Intra-batch Contrast (No Bank) | 90.2 |
| Inter-batch Contrast | **90.8** |

Table 4: Comparison of feature and graph contrast.

| Model (CTR-GCN) | Acc (%) |
|---|---|
| Baseline | 89.8 |
| Feature Contrast | 90.2 |
| Graph contrast | **90.8** |

Table 5: Impact of memory banks.

| Model (CTR-GCN) | Acc (%) |
|---|---|
| Baseline | 89.8 |
| Instance Memory | 90.3 |
| Semantic Memory | 90.2 |
| Instance + Semantic | **90.8** |

Table 6: Impact of sampling strategies. R: Random; H: Hard.

| Sampling | | Acc(%) |
|---|---|---|
| Positive | Negative | |
| | R | 90.1 |
| R | H | 90.2 |
| | R+H | 90.5 |
| | R | 90.4 |
| H | H | 90.6 |
| | R+H | **90.8** |
| R + H | R + H | 90.3 |

Table 7: Comparison with using triplet loss.

| Model (CTR-GCN) | Acc (%) |
|---|---|
| Baseline | 89.8 |
| Triplet loss | 90.7 |
| InfoNCE loss | **90.8** |

Table 8: Training consumption on NTU60.

| Model | Time (hours) |
|---|---|
| 2S-AGCN | 3.4 |
| 2S-AGCN *w*/ours | 3.5 ↑ 2.9% |
| CTR-GCN | 11.4 |
| CTR-GCN *w*/ours | 11.7 ↑ 2.6% |
| InfoGCN | 4.3 |
| InfoGCN *w*/ours | 4.6 ↑ 7.0% |

Table 9: Graph Distance (dis.) comparison using Euclidean distance ($10^{-5}$).

| Sample | Average dis. to all classes | Dis. to correct class | Dis. to misclassified class |
|---|---|---|---|
| Incorrectly-Classified Samples | 1.70 | 0.74 | 0.68 |
| Correctly-Classified Samples | 2.20 | 0.47 | - |

Table 10: Performance (%) of samples with different graph distance ranks to the correct class.

| | CTR | CTR *w/Ours* | +/- |
|---|---|---|---|
| rank1 | 97.2 | 97.4 | +0.2 |
| rank2-5 | 92.1 | 92.9 | +0.8 |
| rank6-10 | 88.3 | 89.3 | +1.0 |
| rank11-20 | 84.9 | 86.3 | +1.4 |
| rank21-40 | 83.2 | 84.8 | +1.6 |
| rank41-60 | 80.8 | 82.9 | +2.1 |

**Intra-batch vs. Inter-batch Graph Contrast.** In Tab. 3, the effectiveness of introducing cross-sequence context is investigated. We find that only contrasting the graphs within one batch can bring improvement with 0.4% (89.8% to 90.2%), which owes to the cross-sequence relation mining. And further exploring the inter-batch relations can bring more improvements to 1.0% (89.8% to 90.8%), which explains that different batches provide richer context than a single batch.

**Graph Contrast vs. Feature Contrast.** In Tab. 4, the comparison of using features $f$ to contrast and using graphs $g$ to contrast is investigated. We find that feature contrast can improve the performance on the baseline with 0.4% (89.8% to 90.2%). But graph contrast can obviously outperform it by 0.6% (90.2% to 90.8%). The results suggest that, due to the high-order structural information in graphs, graph contrast can better benefit graph convolution learning in GCNs.

**Memory Banks.** In Tab. 5, the effectiveness of instance-level and semantic-level memory banks is investigated. We find that both memory banks benefit the recognition, and using them together achieves much higher performance, which proves their complementary properties.

**Sampling Strategy.** In Tab. 6, we compare different sampling strategies for SkeletonGCL. We find that selecting hard positive/negative examples can generally improve recognition. And also random negative samples are meaningful to recognition, which allows the contrastive loss to involve more negative samples.

**InfoNCE Loss vs. Triplet Loss.** In Tab. 7, we compare the performance of using another popular metric learning loss, *i.e.,* triplet loss (Schroff et al., 2015). We find that using triplet loss can achieve similar performance compared to InfoNCE loss. The results indicate the generality of our idea that it does not depend on a certain loss but can boost the performance using different losses.

**Traning Comsumption.** In Tab. 8, we report the training consumption on NTU60. With our method, the training time only slightly increases with different baseline models, ranging from 2.6% to 7.0%, which proves the efficiency of the design.

**Quantitative Results about Graph Similarities.** As shown in Tab. 9, we statistically calculate the graph distances between each sample and all classes (detailed in App. 6.4). For incorrectly-classified samples, we find that: (1) The graph distance to the misclassified class (0.68) is much lower than the average distance (1.70) to all classes. (2) The graph distance to the misclassified class (0.68) is indeed slightly lower than the distance to the correct class (0.74), which explains that not learning class-specific graphs could truly degrade recognition performance. In addition, for correctly-classified samples, we notice that: (1) The average graph distance (2.20) is higher than that for the misclassified ones (1.70), which indicates that the inter-class graph representations are more dispersed for the correctly-classified samples. (2) The distance to the correct class (0.47) is lower than that for the misclassified ones (0.74), which reveals that the intra-class representations are more compact for the correctly classified samples. To sum up, these quantitative results illustrate the strong correlation between recognition performance and class-specific graph representation.

**Performance vs. Graph Quality.** In Tab. 10. we first calculate the graph distances between each sample and all classes (detailed in App. 6.4) for CTR-GCN. Then, we rank the distances from low to high. In Tab. 10, we report the recognition accuracies of samples according to their distance ranks to the correct class. Here, higher ranks indicate that graphs are of higher quality (intra-class compact and inter-class dispersed), while lower ranks indicate that graphs are of lower quality (intra-class dispersed and inter-class aliasing). We note that: (1) Considering samples from lower ranks to higher ranks, performances improve monotonically, revealing the significant correlations between graph quality and recognition performance. (2) Combined with the proposed method, we improve performances in all cases, where larger improvements are obtained on the samples with lower-quality graphs. These results prove that our method can alleviate the problem caused by learning low-quality graphs.

## 5 CONCLUSION

In this paper, we establish a new training paradigm for skeleton-based action recognition, called SkeletonGCL, which explicitly explores the rich semantic context across sequences. Concretely, SkeletonGCL contrasts the learned graphs among sequences, guiding the graph representations to be class-associated, hence improving GCN capacity to recognize different actions. We improve the current methods significantly to achieve SoTA on three benchmarks.

**Limitation.** In this paper, we push away the negative pairs from different classes in the same way without considering their intrinsic relations. Therefore, a comprehensive contrasting manner may be more suitable by delicately involving cross-class relations. We leave this for future work.

## ACKNOWLEDGEMENTS

This research is supported by the NSFC (grants No. 61773176 and No. 61733007).

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

# 6 APPENDIX

## 6.1 IMPLEMENTATIONS OF GRAPH PROJECTION HEADS FOR GCNS.

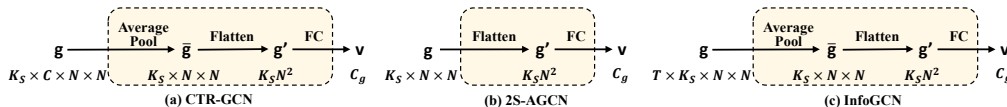

Figure 3: Illustration of graph projection heads for GCNs.

In Fig. 3, we illustrate the implementation details of graph projection heads for different GCNs. Particularly, for CTR-GCN and InfoGCN, we first apply an average pooling layer to summarize the information along the channel and temporal dimensions, respectively. Then, the same as 2S-AGCN, we flatten the graphs and embed them with an FC layer.

## 6.2 MORE DIAGNOSTIC EXPERIMENTS.

Table 11: Comparison of using cross-entropy loss to supervise graph learning.

| Model (CTR-GCN) | Acc (%) |
|---|---|
| Baseline | 89.8 |
| *w/*Cross-entropy loss | 89.7 |
| *w/*SkeletonGCL | **90.8** |

Table 12: Impact of the FC layer in graph projection head.

| Model (CTR-GCN) | Acc (%) |
|---|---|
| Baseline | 89.8 |
| Graph Projection *wo/*FC | 90.1 |
| Graph Projection *w/*FC | **90.8** |
| Graph Projection *w/*MLP | 90.6 |

Table 13: Apply graph contrast on different layers.

| Model (CTR-GCN) | Acc (%) |
|---|---|
| Baseline | 89.8 |
| $7th$ layer | 89.8 |
| $8th$ layer | 90.0 |
| $9th$ layer | 90.4 |
| $10th$ layer | **90.8** |

Table 14: Impact of the size of $\mathcal{M}_{\text{Ins}}$.

| $P$ | Acc (%) |
|---|---|
| 128 | 90.3 |
| 342 | 90.5 |
| 684 | **90.8** |
| 1368 | 90.4 |
| 2736 | 90.0 |

Table 15: Performance comparison with different $C_g$.

| $C_g$ | Acc (%) |
|---|---|
| 64 | 90.4 |
| 128 | 90.6 |
| 256 | **90.8** |
| 512 | 90.5 |

Table 16: Impact of $\tau$.

| $\tau$ | Acc (%) |
|---|---|
| 0.5 | 90.4 |
| 0.8 | 90.6 |
| 1.0 | **90.8** |
| 1.5 | 90.5 |

Table 17: Impact of $\alpha$.

| $\alpha$ | Acc (%) |
|---|---|
| 0.75 | 90.6 |
| 0.80 | 90.7 |
| 0.85 | **90.8** |
| 0.9 | 90.6 |

Table 18: Performance comparison with different $K_H^+$.

| $K_H^+$ | Acc (%) |
|---|---|
| 64 | 90.6 |
| 128 | **90.8** |
| 256 | 90.6 |
| 512 | 90.5 |

Table 19: Performance comparison with different $K_H^-$.

| $K_H^-$ | Acc (%) |
|---|---|
| 128 | 90.5 |
| 256 | 90.7 |
| 512 | **90.8** |
| 1024 | 90.5 |

Table 20: Performance comparison with different $K_R^-$.

| $K_R^-$ | Acc (%) |
|---|---|
| 128 | 90.6 |
| 256 | 90.7 |
| 512 | **90.8** |
| 1024 | 90.6 |

**Comparison of using cross-entropy loss.** Since cross-entropy loss is a widely used classification loss in learning class-discriminative representations, in Tab. 11, we investigate its performance to supervise graph learning. We find that directly using cross-entropy loss for graph learning has negligible effects on the performance (89.8% to 89.7%), which indicates that it is impractical to learn favorable class-discriminative graphs by naively using a classification loss. In this paper, we find a practical way to achieve this goal by introducing the cross-sequence context for guiding graph learning.

**Impact of FC in Projection Head.** In Tab. 12, the effectiveness of transformation layer (FC layer) in the graph projection head is investigated. We find that the model achieves obvious improvement (90.1% to 90.8%) equipped with the FC layer, which proves the importance of vertex-aware graph encoding. In addition, we find that using an MLP achieves a similar but lower accuracy, hence we use a simple FC in the framework.

**Which Layer to Contrast Graphs?** In Tab. 13, we apply graph contrast on different layers. We find that contrasting graphs on deeper layers outperform on shallower layers. One possible explanation is that deeper layers can provide higher-level semantics that is relevant to recognition.

**Impact of the size of $\mathcal{M}_{\text{Ins}}$.** In Tab. 14, the impact of the size of $\mathcal{M}_{\text{Ins}}$ is investigated, where we use different values of $P$ to control the size. We find that appropriately increasing the size can effectively expand the cross-sequence context, and improve recognition performance. However, an over large memory bank stores *old* samples from a few batches ago, which hinders representation learning.

**Impact of dimension $C_g$.** In Tab. 15, the influences of the dimension of graph vector $C_g$ are investigated. For pursuing the best performance, we set $C_g$ as 256.

**Impact of temperature $\tau$.** In Tab. 16, the influences of the temperature $\tau$ are investigated. For pursuing the best performance, we set $\tau$ as 1.0.

**Impact of $\alpha$.** In Tab. 17, the influences of the momentum updating hyper-parameter $\alpha$ are investigated. For pursuing the best performance, we set $\alpha$ as 0.85.

**Impact of the number of sampling examples.** In Tab. 18, the impact of selecting $K_H^+$ hardest positive examples is investigated. In Tab. 19, the impact of selecting $K_H^-$ hardest negative examples is investigated. In Tab. 20, the impact of selecting $K_R^-$ random negative examples is investigated. For pursuing the best performance, we set $K_H^+$, $K_H^-$, and $K_R^-$ to 128, 512 and 512, respectively.

**The quantitative analysis of accuracy improvement.** In Tab. 21, the recognition accuracies of the top-10 hardest classes for CTR-GCN on NTU-60 are presented. The improvements in four classes (*i.e.*, "reading", "typing on a keyboard", "headache" and "point to something") are over $4\%$. Though performances in three classes decrease, they are relatively small ($-1.5\%$ on "writing", $-1.0\%$ on "take off a shoe", and $-2.5\%$ on "sneeze/cough") vs. others' increase. Overall, we obtain an average improvement in the 10 classes of $2.7\%$.

In Tab. 22, the recognition accuracies of top-10 improved classes for CTR-GCN on NTU-60 are presented. The accuracy of the above 10 classes shows an average gain of $4.6\%$.

Table 21: Performance (%) on top 10 hardest classes for CTR-GCN.

| classes | CTR-GCN | CTR-GCN w/GCL | +/- |
|---|---|---|---|
| reading | 58.2 | 67.8 | +9.6 |
| type on a keyboard | 66.5 | 70.9 | +4.4 |
| writing | 67.3 | 65.8 | -1.5 |
| play with phone/tablet | 68.0 | 70.9 | +2.9 |
| eat meal | 71.6 | 75.3 | +3.7 |
| take off a shoe | 75.5 | 74.5 | -1.0 |
| headache | 78.3 | 83.0 | +4.7 |
| point to something | 81.9 | 85.9 | +4.0 |
| clapping | 82.1 | 85.7 | +3.6 |
| sneeze/cough | 82.2 | 79.7 | -2.5 |
| Average | 73.2 | 75.9 | +2.7 |

Table 22: Performance (%) on top 10 improved classes for CTR-GCN.

| classes | CTR-GCN | CTR-GCN w/GCL | +/- |
|---|---|---|---|
| reading | 58.2 | 67.8 | +9.6 |
| headache | 78.3 | 83.0 | +4.7 |
| rub two hands | 87.3 | 92.0 | +4.7 |
| punch/slap | 89.4 | 93.8 | +4.4 |
| type on a keyboard | 66.6 | 70.9 | +4.3 |
| point to something | 81.9 | 85.9 | +4.0 |
| clapping | 82.1 | 85.7 | +3.6 |
| reach into pocket | 82.5 | 86.1 | +3.6 |
| eat meal | 71.6 | 75.2 | +3.6 |
| neck pain | 88.0 | 91.3 | +3.3 |
| Average | 78.6 | 83.2 | +4.6 |

## 6.3 QUALITATIVE RESULTS

In Fig. 4, we visualize the t-SNE distribution of graph and feature representations of sequences from six classes, illustrating the impact of SkeletonGCL. As shown in Fig. 4(a), SkeletonGCL can shape the graph representation structure, where the graphs from the same class get together and graphs from different classes spread out. Consequently, in Fig. 4(b), with SkeletonGCL, the features from different classes become more distinguishable, which indicates that graph contrast indeed improves the feature extraction capacity.

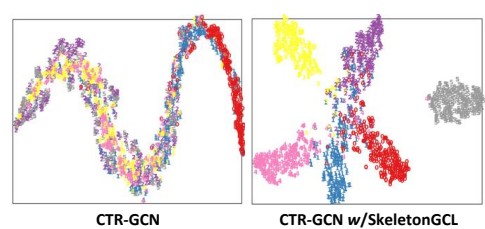
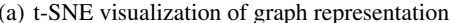

(a) t-SNE visualization of graph representation.

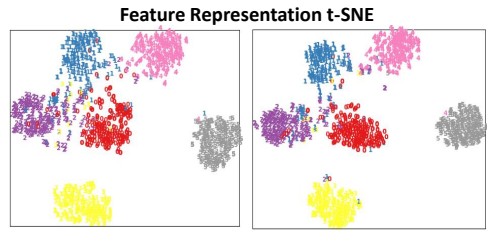

(b) t-SNE visualization of feature representation.

Figure 4: **t-SNE visualization.** t-SNE (Van der Maaten & Hinton, 2008) visualization of graph and feature representations from sequences in the test set of NTU 60. Each color denotes a certain class. Best viewed with zoom in.

## 6.4 THE CALCULATION DETAILS OF GRAPH DISTANCE

In Tab. 9, the statistics of graph distance for all samples are investigated. In CTR-GCN, the graph $\mathbf{g} \in \mathbb{R}^{K_S \times C \times N \times N}$ is learned for graph convolution. For the convenience of calculation, we use an average pooling to squeeze $\mathbf{g}$ and reshape it into $\boldsymbol{g} = \overline{\mathbf{g}} \in \mathbb{R}^{1 \times N^2}$ as the graph embedding to conduct distance measurement.

To acquire the graph embedding for each class $c_*$, we first calculate the centroid vector $s$ as follows,

$$\boldsymbol{s}_{c_*} = \frac{1}{N_{c_*}} \sum_{i=1}^{N_{c_*}} \boldsymbol{g}_i \tag{9}$$

where $N_{c_*}$ denotes the number of samples which belongs to class $c_*$. To effectively reveal the relation between graph quality and classification accuracy, we introduce three types of graph distance, i.e., $d_{\mathrm{all}}$, $d_{\mathrm{cor}}$ and $d_{\mathrm{mis}}$. $d_{\mathrm{all}}$ measures the average distance to all classes. $d_{\mathrm{cor}}$ measures the distance to the correct class $c_*$. $d_{\mathrm{mis}}$ measures the distance to the misclassified class $c_{\mathrm{mis}}$.

$$d_{\mathrm{all}} = \frac{1}{K} \sum_{k=1}^{K} ||\boldsymbol{g} - \boldsymbol{s}_{c_k}||^2 \tag{10}$$

$$d_{\mathrm{cor}} = ||\boldsymbol{g} - \boldsymbol{s}_{c_*}||^2, \tag{11}$$

$$d_{\mathrm{mis}} = ||\boldsymbol{g} - \boldsymbol{s}_{c_{\mathrm{mis}}}||^2, \tag{12}$$

where $K$ denotes the number of classes in the dataset.

