# OpenReview forum: "Graph Contrastive Learning for Skeleton-based Action Recognition"
_ICLR.cc/2023/Conference — ICLR 2023 poster_

### Official Review · Reviewer_Cq4q · 2022-10-23

**Confidence:** 4
**Correctness:** 4
**Technical Novelty And Significance:** 2
**Empirical Novelty And Significance:** 2
**Recommendation:** 6

**Clarity, Quality, Novelty And Reproducibility:**

**Clarity**

1. It seems that the memory banks are a part of the authors' main contributions. But they are not listed as part of the main contributions (end of Sec. 1).

2. I found the use of subscripts and superscripts to indicate positives and negatives inconsistent. For example, in Sec. 3.1, the authors write positive input samples and features as $I^+$ and $f^+$, but write their negative set as $\mathcal{N}_-$ (more such examples in the "Loss" and "Hard Sampling" paragraphs). If there is any specific reasoning behind these differences, I would be happy to learn it.

3. I would recommend explicitly writing down the loss function for $\mathcal{L}_\textrm{NCE}^\textrm{Sem}$ for completeness.


**Quality and Novelty**

Without any additional insight into the results (please refer to my concerns under "weaknesses"), the contributions of the proposed method appear marginal, with most of the heavy lifting already done by the current graph convolutional methods. Also, explaining the differences between aggregating the instance-level memory bank features and using the semantic-level memory banks more rigorously can help in understanding the quality of the contributions.


**Reproducibility**

The authors sufficiently describe their proposed components so their method is reproducible.

**Strength And Weaknesses:**

**Strengths**
1. The proposed approach of using contrastive learning and the memory bank descriptions are technically sound.
2. The paper is easy to follow overall and well-organized.


**Weaknesses**

My main concern is regarding the results, particularly in the net improvement the authors' proposed techniques achieve.

1. All the experiments show a 1% or less improvement in accuracy, but the authors haven't discussed whether such improvements are significant. Assuming these are the mean accuracy numbers, what are the corresponding standard deviations?

2. Do the authors have confusion matrices to show the intra-class and inter-class performances? Given that the baseline accuracies are already close to 90%, it can perhaps help to show how much the proposed method improves performance just on hard data samples (note that this is different from the performance improvement using hard sampling that the authors report in Table 6).

3. Somewhat along these lines, I see visual evaluations (Fig. 3) that provide a better perspective of how contrastive learning helps. Can the authors further explain why the graph representation changes dramatically (Fig. 3a) while the feature representations stay almost the same (Fig. 3b)?

4. For Table 5, have the authors performed any additional experiments with the instance-level memory bank size $P$?

5. I am also not fully clear on the need for a separate semantic-level memory bank that aggregates the graph-based features across all intra-class samples. How different is that from aggregating the features in the instance-level memory bank per class? I can see that the memory bank size $P$ can play a role here, how large is the value of $P$ compared to the class populations?

6. Further, how does the hard sampling work in conjunction with the FIFO setup of the instance-level memory banks? Which samples are removed from the queue after the similarity calculation?

**Summary Of The Paper:**

The authors present a method to perform skeleton-based action recognition using contrastive learning with graph sequences. To aid in contrastive learning, the authors propose two memory banks to store the graph-based data: an instance-level memory bank to store the properties of individual data samples and a semantic-level memory bank to store the aggregate properties of all the samples in each action class. The combined implementation of the two memory banks enables their learned network to cluster together intra-class samples and spread out inter-class samples. The authors integrate their contrastive learning approach with multiple graph convolutional techniques for skeleton-based action recognition and experimentally demonstrate the benefits of their proposed components.

**Summary Of The Review:**

The authors tackle the well-known problem of skeleton-based action recognition that has mature solutions available, and propose a contrastive learning-based method that integrates with the current solutions and numerically offers only marginal benefits. While that by itself does not discount the importance of their contribution, I recommend the authors describe the differences between their memory banks more rigorously and better explain where and how their proposed method performs best to get a clearer understanding of the quality of their work.

---

> ### Author Response · Authors · 2022-11-19
> **Response to Reviewer Cq4q (Part 1/3)**
>
> Thanks for your precious time reading this paper. We are delighted that you found the paper technically sound and easy to follow. And your constructive suggestions are of great help to us.
>
> Please refer to the detailed response below, regarding the concerns you raised.
>
> >**Q1:** Whether such improvements are significant?
>
> **R:** If comparing to the performance improvements made by previous state-of-the-art methods, the improvements brought by our SkeletonGCL are significant. 2S-AGCN (CVPR2019), CTR-GCN (ICCV2021) and InfoGCN (CVPR2023) are all SoTA methods when publishing. In Tab. 1, the improvements among them from 2019 to 2022 on NTU-based settings are between 0.2~1.4. In Tab. 2, it is notable that the lifting on performance with our method is consistent for all three popular GCN-based methods. Considering NTU60 and NTU120 are extensively benchmarked, such consistent improvements are hard-won and become more significant.
>
> **Table 1: Performance improvement (%) brought by previous SoTA methods.**
> | Method       | Track    | NTU60-XSub-J | NTU60-XView-J | NTU120-XSub-J | NTU120-XSet-J |
> |--------------|----------|------------|------------|-------------|-------------|
> | 2S-AGCN      | CVPR2019 | 88.9          | 94.5          | 84.0           | 85.3           |
> | CTR-GCN      | ICCV2021 | 89.8 ($\uparrow$$0.9$)       | 94.8 ($\uparrow$$0.3$)       | 84.9 ($\uparrow$$0.9$)        | 86.7 ($\uparrow$$1.4$)        |
> | InfoGCN      | CVPR2022 | 90.3 ($\uparrow$$0.5$)       | 95.5 ($\uparrow$$0.7$)       | 85.1 ($\uparrow$$0.2$)        | 86.3 ($\downarrow$$0.4$)        |
>
> **Table 2: Performance improvement (%) brought by our method.**
> | Method       | Track    | NTU60-XSub-J | NTU60-XView-J | NTU120-XSub-J | NTU120-XSet-J
> |--------------|----------|------------|------------|-------------|-------------|
> | 2S-AGCN      | CVPR2019 | 88.9       | 94.5       | 84.0        | 85.3        |
> | +SkeletonGCL |     -    | 89.9 ($\uparrow$$1.0$)       | 95.0 ($\uparrow$$0.5$)       | 84.6 ($\uparrow$$0.6$)       | 86.1 ($\uparrow$$0.8$)        |
> | CTR-GCN      | ICCV2021 | 89.8       | 94.8       | 84.9        | 86.7        |
> | +SkeletonGCL |     -    | 90.8 ($\uparrow$$1.0$)      | 95.3 ($\uparrow$$0.5$)       | 85.6 ($\uparrow$$0.7$)        | 97.3 ($\uparrow$$0.6$)        |
> | InfoGCN      | CVPR2022 | 89.4       | 95.2       | 84.2        | 86.3        |
> | +SkeletonGCL |     -    | 90.1 ($\uparrow$$0.7$)       | 95.5 ($\uparrow$$0.3$)       | 85.2 ($\uparrow$$1.0$)        | 87.2 ($\uparrow$$0.9$)        |
>
> It is noted that all the reported results are reproduced with their officially released code under the same experimental settings.
>
> > **Q2:** what are the standard deviations of performances?
>
> **R:** Taking the case of CTRGCN on the NTU60 dataset, we change the random seed to different values and run the experiments five times. Tab.3 shows the mean accuracy and standard deviation:
>
> **Table 3: Performance comparison with standard deviation.**
> | Model | Accuracy (%) |
> | -------- | -------- |
> | CTRGCN     |   89.76 $\pm$ 0.07  |
> | CTRGCN *w/SkeletonGCL* | 90.77 $\pm$ 0.06 |
>
> From the results, we can notice that the results are rather stable.

---

> ### Author Response · Authors · 2022-11-19
> **Response to Reviewer Cq4q (Part 2/3)**
>
> > **Q3:** Do the authors have confusion matrices to show the intra-class and inter-class performances? How much the proposed method improves performance on hard data samples?
>
> **R:** Thanks for your suggestion. We have included the visualizations of confusion matrices in the *supplementary material* of the revision. However, since the performance is already close to 90%, we find that such visualizations are insufficient to explain our improvement effectively. To help understand the improvements more clearly, we provide comprehensive analyses below, by which we hope to alleviate your concerns.
>
> In Tab. 4, the recognition accuracies of the top 10 hardest classes for CTR-GCN on NTU-60 are presented. The improvements in four classes (i.e., "reading", "typing on a keyboard", "headache" and "point to something") are over 4%. Though performances in three classes decrease, they are relatively small (-1.5% on "writing", -1.0% on "take off a shoe", and -2.5% on "sneeze/cough") vs. others' increase. Overall, we obtain an average improvement in the 10 classes of 2.7%.
>
> **Table 4: Performance (%) on top 10 hardest classes for CTR-GCN**
> | classes                | CTR-GCN | CTR-GCN w/GCL | +/-  |
> |------------------------|---------|---------------|-------|
> | reading                | 58.2    | 67.8          | +9.6   |
> | type on a keyboard     | 66.5    | 70.9          | +4.4   |
> | writing                | 67.3    | 65.8          | -1.5  |
> | play with phone/tablet | 68.0    | 70.9          | +2.9   |
> | eat meal               | 71.6    | 75.3          | +3.7   |
> | take off a shoe        | 75.5    | 74.5          | -1.0  |
> | headache               | 78.3    | 83.0          | +4.7   |
> | point to something     | 81.9    | 85.9          | +4.0   |
> | clapping               | 82.1    | 85.7          | +3.6   |
> | sneeze/cough           | 82.2    | 79.7          | -2.5  |
> Average |               73.2    | 75.9          | +2.7 |
>
> In Tab. 5, the recognition accuracies of top 10 improved classes for CTR-GCN on NTU-60 are presented. The accuracy of the above 10 classes shows an average gain of 4.6%.
>
> **Table 5: Performance (%) on top 10 improved classes for CTR-GCN**
> | classes            | CTR-GCN | CTR-GCN w/GCL | +/- |
> |--------------------|---------|---------------|------|
> | reading            | 58.2    | 67.8          | +9.6  |
> | headache           | 78.3    | 83.0          | +4.7  |
> | rub two hands      | 87.3    | 92.0          | +4.7  |
> | punch/slap         | 89.4    | 93.8          | +4.4  |
> | type on a keyboard | 66.6    | 70.9          | +4.3  |
> | point to something | 81.9    | 85.9          | +4.0  |
> | clapping           | 82.1    | 85.7          | +3.6  |
> | reach into pocket  | 82.5    | 86.1          | +3.6  |
> | eat meal           | 71.6    | 75.2          | +3.6  |
> | neck pain          | 88.0    | 91.3          | +3.3  |
> | Average | 78.6 | 83.2 | +4.6 |
>
>
> Besides, we provide another in-depth analysis to present our improvements on hard examples. Specifically, we calculate graph distances between each sample and all classes (details in App. 6.3). Then, we rank the distances from low to high. In Tab. 6, we report the recognition accuracies of samples according to their distance ranks to the ground-truth class. Here, higher ranks indicate that graphs are of higher quality (intra-class compact and inter-class dispersed), while lower ranks indicate that graphs are of lower quality (intra-class dispersed and inter-class aliasing).
>
> **Table 6: Performance (%) of samples with different graph distance ranks to the correct class.**
> |          | CTR   | CTR w/Ours | +/-  |
> |----------|-------|------------|-------|
> | rank1     | 97.2  | 97.4       | +0.2  |
> | rank2-5   | 92.1  | 92.9       | +0.8  |
> | rank6-10  | 88.3  | 89.3       | +1.0  |
> | rank11-20 | 84.9  | 86.3       | +1.4  |
> | rank21-40 | 83.2  | 84.8       | +1.6  |
> | rank41-60 | 80.8  | 82.9       | +2.1  |
>
> Several noteworthy observations are: (1) Considering samples from lower ranks to higher ranks, performances improve monotonically, revealing the significant correlations between graph quality and recognition performance. In this case, samples with low-quality graphs can be seen as **hard examples**. (2) Combined with the proposed method, we improve performances in all cases, where larger improvements are obtained on the **harder samples** with lower-quality graphs. These results prove that our method can alleviate the problem caused by low-quality graph learning, which have been included in the revision (Tab. 10 of Sec. 4.4).

---

> ### Author Response · Authors · 2022-11-19
> **Response to Reviewer Cq4q (Part 3/3)**
>
> > **Q4:** Somewhat along these lines, I see visual evaluations (Fig. 3) that provide a better perspective of how contrastive learning helps. Can the authors further explain why the graph representation changes dramatically (Fig. 3a) while the feature representations stay almost the same (Fig. 3b)?
>
> **R:** The graphs and the features are in different embedding spaces. If we take a closer look at Figure 3-(b), it is found that the boundaries between different classes with GCL are more clear than those without GCL. It shows that our SkeletonGCL can effectively help the GCN-based models deal with those ambiguous samples or hard samples.
>
> >**Q5:** Have the authors performed any additional experiments with the instance-level memory bank size $P$?
>
> **R:** We indeed perform additional experiments with different bank sizes $P$ in Tab. 7 below.
>
> **Table 7: Performance comparison with different $P$.**
> | $P$ | Acc (%) |
> | -------- | -------- |
> | 128 | 90.3 |
> | 342 | 90.5 |
> | **684** | **90.8** |
> | 1368 | 90.4 |
> | 2736 | 90.0 |
>
> From the results, we can notice that increasing $P$ can effectively expand the cross-sequence context and improve recognition performance. However, an over large memory bank stores *old* samples from a few batches ago, which hinders representation learning.
>
> >**Q6:** The need for a semantic-level memory bank and the value of P compared to class populations.
>
> **R:** The instance-level memory bank and the semantic-level memory bank provides different views for contrastive learning. Specifically, the former provides instances that keep **individual characteristics** of sequences to enrich diversities, while the latter provides instances that contain class-level representations standing for **general properties** of each class. In this way, the instance-level memory bank emphasizes **individuality**, while the semantic-level memory bank pays attention to **generality**, which are complementary to each other. Tab. 8 shows that using them together achieves much higher performance, which proves their complementary functions.
>
> **Table 8: Impact of memory banks.**
> | Models            | Acc(%) |
> |-------------------|--------|
> | Baseline          | 89.8   |
> | Instance Memory   | 90.3   |
> | Semantic Memory   | 90.2   |
> | Instance+Semantic | **90.8**   |
>
> The value $P$ is set through an ablation study as shown in Tab. 7. Interestingly, we find that the best $P$ (684) is close to class populations (about 685 samples per class).
>
> > **Q7:** Further, how does the hard sampling work in conjunction with the FIFO setup of the instance-level memory banks? Which samples are removed from the queue after the similarity calculation?
>
> **R:** We elaborate on how hard sampling works with the FIFO setup in the instance-level memory bank below.
>
> (1) Given a sequence $X$ with class $c_*$, the GCN encoder and graph projection head produce a graph embedding $v \in \mathbb{R}^{C_g}$.
>
> (2) Given an instance-level memory bank $\mathcal{M}_\text{Ins} \in \mathbb{R}^{C_k \times P \times C_g}$, we calculate the cosine similarities between $v$ and all instances in  $\mathcal{M}_\text{Ins}$.
>
> Note, instances from class $c_*$  are regarded as positive instances and the others are regarded as negative instances.
>
> (3) Then, based on the values of calculated similarities, we select $K_H^+$ hardest positive instances, $K_H^+$ hardest negative instances, and $K_R^+$ random negative instances, to conduct contrastive loss $\mathcal{L}_\text{NCE}^\text{Ins}$.
>
> (4) Afterward, we update the queue in the memory bank for class $c_*$ by appending the embedding $v$ at its rear and poping out its head instance.
>
> Therefore, no samples will be removed from the queue after the similarity calculation. Removing the head instance only occurs after calculating loss.
>
> > **Q8:** It seems that the memory banks are a part of the authors' main contributions. But they are not listed as part of the main contributions (end of Sec. 1).
>
> **R:** Thanks for your appreciation of the contributions of our memory bank design. We follow your suggestion and have included it as part of the main contribution in the revision (end of Sec. 1).
>
> > **Q9:** The suggestions on notations.
>
> **R:** Thanks for your kind suggestions and we have revised them in the revision.
>
> > **Q10:** I would recommend explicitly writing down the loss function for completeness.
>
> **R:** Thanks for your careful review. We have included the corresponding formulation in the revision (Eq. (6) of Sec. 3.).
>
> If the response satisfies your concerns, please consider reassessing the score. Thanks for your valuable comments and looking forward to your reply.

---

> ### Comment · Reviewer_Cq4q · 2022-11-22
> **Reviewer Response**
>
> Thank you for the very detailed response, it addresses all my questions and most of my concerns. While I am still not fully convinced that the overall performance gains of ~1% are significant enough by themselves, I can appreciate the additional performance reports, especially the overall standard deviation (which appears quite stable), and the more significant performance gains on the hardest classes. For these reasons, I am updating my recommendation to "above the acceptance threshold".

---

> > ### Author Response · Authors · 2022-11-25
> > **Response to Reviewer Cq4q**
> >
> > Thanks for expressing your appreciation for the additional performance reports and giving a positive assessment of this work. It means a lot to us.

---

### Official Review · Reviewer_vLtp · 2022-10-24

**Confidence:** 3
**Correctness:** 4
**Technical Novelty And Significance:** 3
**Empirical Novelty And Significance:** 3
**Recommendation:** 8

**Clarity, Quality, Novelty And Reproducibility:**

The paper is well written. The main contribution of this paper is the new training paradigm – graph contrastive learning, which can be integrated into other GCNs. It would be great if the authors can release their code. It may be hard to reproduce the results

**Strength And Weaknesses:**

Strength:

It proposes the graph contrastive learning framework for action recognition.  This framework can be integrated into other graph convolution networks. The experimental results have shown that the proposed framework is integrated into three GCNs and is able to improve performance.

Weaknesses:

Regarding the instance-level memory bank, how to assign the number of instances to different classes? It would be great to add more descriptions. The infoNCE loss for the semantic part should be provided since this part is novel.

**Summary Of The Paper:**

This paper proposes a graph contrastive learning framework to explore the global context across all sequences for action recognition. It aims to enforce graphs to be class-discriminative.  More importantly, it proposes two memory banks for two complementary levels: instance and semantic levels. This method can be integrated into other graph convolution networks (GCN). The experimental results on three public datasets are better than other compared approaches.

**Summary Of The Review:**

The graph contrastive learning is novel. In addition, it could be integrated into other GCNs and improves the performance of action recognition.

---

> ### Author Response · Authors · 2022-11-19
> **Response to Reviewer vLtp**
>
> Thanks for your precious time reading this paper. We are delighted that you found the paper well-written and novel.
>
> Please refer to the detailed response below about several weaknesses you raised.
>
> >**Q1:** Regarding the instance-level memory bank, how to assign the number of instances to different classes?
>
> **R:** Since the number of sequences in different classes is relatively close to each other in datasets, *e.g.*, NTU60, we assign the same number of instances ($P$) for different classes in the instance-level memory bank. We set the value of $P$ on the NTU60 dataset based on the ablation study in Tab. 1. From the results, we can notice that increasing $P$ can effectively expand the cross-sequence context and improve recognition performance. However, an over large memory bank stores *old* samples from a few batches ago, which hinders representation learning. Interestingly, we find that the best $P$ (684) is close to class populations (about 685 samples per class).
>
> **Table 1: Performance comparison with different $P$.**
> | $P$ | Acc (%) |
> | -------- | -------- |
> | 128 | 90.3 |
> | 342 | 90.5 |
> | **684** | **90.8** |
> | 1368 | 90.4 |
> | 2736 | 90.0 |
>
> > **Q2:** The infoNCE loss for the semantic part should be provided since this part is novel.
>
> **R:** Thanks for your careful review. We have included the corresponding formulation in the revision (Eq. (6) of Sec. 3.).
>
> > **Q3:** It would be great if the authors can release their code. It may be hard to reproduce the results
>
> **R:** We promise that we will publish the source code to facilitate the development of the research community.
>
> We hope the explanations and results above solve your concerns. Thanks for your valuable comments and looking forward to your reply.

---

### Official Review · Reviewer_YfcM · 2022-10-24

**Confidence:** 5
**Correctness:** 2
**Technical Novelty And Significance:** 2
**Empirical Novelty And Significance:** 2
**Recommendation:** 5

**Clarity, Quality, Novelty And Reproducibility:**

The paper is clearly written and I believe it can be reproduced from the description of the method.
The novelty is limited: on the technical side, any existing GCN is combined with a contrastive loss on the graphs. On the conceptual side, a novel claim is made but unfortunately is not backed by evidence.

**Strength And Weaknesses:**

**Strengths**

(a) technical idea
- using two memory banks to explicitly control an instance-based and a semantic loss part is a simple and effective solution
- the idea to explicitly learn graph contrasts to avoid similar graphs that lead to classification error is, to the best of my knowledge, novel

(b) results
- several recent state of the art benchmarks are consistently outperformed
- the authors provide a fair comparison, where they report the original numbers from the referenced papers, but also retrain those methods and report the retrained results, although they are frequently better than the numbers reported in the original paper. I appreciate the care with which results are presented.
- the t-SNE plot shows that the proposed method leads to clearer clusterings of produced graphs, which was the goal of the paper.

**Weaknesses**

(a) Novelty
The proposed method is essentially any existing graph convolutional method trained with an additional contrastive loss. While it is interesting to see the improvements this causes, the technical novelty of this approach is rather limited.

(b) Motivation and Evaluation
The authors motivate their approach by the hypothesis "From the visualization, we find that (1) For sequences that are
correctly classified in Fig. 1 (a) and Fig. 1 (b), the learned graphs in the same class look similar,
while graphs in different classes have distinct differences. (2) For a misclassified sequence in Fig.
1 (c), the learned graph resembles the graphs from the misclassified class more than those from the
ground truth class."
Unfortunately, they do not provide quantitative evidence of this hypothesis. While there is one qualitative example shown in Fig. 1, this is not evidence that the problem is a broad problem for GCN based action recognition. The hypothesis needs to be backed by some quantitative metrics, e.g. showing that misclassifications strongly correlate with graph distances to the misclassified class being lower than average or lower than to the correct class.

(c) Results
Similar as for the motivation, the results are lacking evidence that the proposed method alleviates the problem that is claimed to cause lower performance, i.e., the problem that misclassified graphs are too similar to graphs from other classes.
While the proposed approach leads to some improvement over state of the art methods, it is unclear if this improvement is caused by improved graph structures as claimed/motivated by the authors.
An in-depth analysis of the main claim of the paper is therefore missing.

**Summary Of The Paper:**

The authors identify the lack of global context as a limiting factor of current state of the art, graph-convolution-based methods for skeletal action recognition. They propose a training paradigm based on a contrastive loss that is applicable to most existing approaches. Their proposed method is evaluated against existing work on the most widely used benchmarks and an ablation study investigating the impact of different model components is provided.

**Summary Of The Review:**

The paper presents some improvements over state of the art in skeleton-based action recognition. The main claim of the paper (i.e., that graph structure and their lack of contrast to the misclassified class is responsible for limited performance of existing works,) is not backed by quantitative evidence. Similarly, the results are lacking an analysis if the proposed contrastive learning scheme solves the proposed problem. Obviously there is improvement, but there is no convincing analysis if the improvement roots in better graph structures.

---

> ### Author Response · Authors · 2022-11-19
> **Response to Reviewer YfcM (Part 1/2)**
>
> Thanks for your precious time reading this paper. We are delighted that you found the paper well-written. And your constructive suggestions are of great help to us.
>
> Please refer to our detailed response below, regarding your concerns about novelty, quantitative evidence of the motivation hypothesis, and performance analysis.
>
> >**Q1:** The proposed method is essentially any existing graph convolutional method trained with an additional contrastive loss. While it is interesting to see the improvements this causes, the technical novelty of this approach is rather limited.
>
> **R:** Our work is far more than a simple mix of GCN and contrastive learning. In this work, we propose graph contrastive learning for skeleton-based action recognition (SkeletonGCL), involving cross-sequence context to learn class-specific graphs (intra-class compact and inter-class dispersed), different from all previous methods that only use intra-sequence features to learn graphs, which do not explicitly associate with class semantics. Technically, we design two memory banks and a series of techniques dedicatedly (e.g., graph projection and hard sampling) to make the above idea work, which play crucial roles as shown in the experiments. Besides, to validate the feasibility of SkeletonGCL, plenty of ablation studies are conducted and presented with trial and error. In the end, SkeletonGCL achieves consistent improvements with three GCNs on three popular datasets under multiple testing protocols, which strongly validates the efficiency and generalization of our idea.
>
> >**Q2:** Unfortunately, they do not provide quantitative evidence of this hypothesis. The hypothesis needs to be backed by some quantitative metrics, e.g. showing that misclassifications strongly correlate with graph distances to the misclassified class being lower than average or lower than the correct class.
>
> **R:** Thanks for your nice suggestion. We have included quantitative results and analysis to further support our hypothesis in the revision (Tab. 9 of Sec. 4.4). As shown in Tab. 1, we take CTRGCN as the feature extractor and we statistically calculate the graph distances between each sample and all classes (detailed in App. 6.3) on the NTU60 dataset. For incorrectly-classified samples, we find that: (a) The graph distance to the misclassified class (0.68) is much lower than the average distance (1.70) to all classes. (b) The graph distance to the misclassified class (0.68) is indeed slightly lower than the distance to the correct class (0.74), which explains that not learning class-specific graphs could truly degrade recognition performance.
>
> **Table 1: Graph distance comparison using Euclidean distance ($10^{-5}$).**
> | Sample                         | Average Distance to All Classes | Distance to Correct class | Distance to Misclassified Class |
> |--------------------------------|---------------------------------|----------------------|---------------------------------|
> | Incorrectly-Classified Samples | 1.70                            | 0.74                 | 0.68                            |
> | Correctly-Classified Samples   | 2.20                            | 0.47                 | -                               |
>
> In addition, for correctly-classified samples, we notice that: (a) The average graph distance (2.20) is higher than that for the misclassified ones (1.70), which indicates that the inter-class graph representations are more dispersed for the correctly-classified samples. (b) The distance to the correct class (0.47) is lower than that for the misclassified ones (0.74), which reveals that the intra-class representations are more compact for the correctly-classified samples.
>
> To sum up, these quantitative results illustrate the strong correlation between recognition performance and class-specific graph representation.

---

> ### Author Response · Authors · 2022-11-19
> **Response to Reviewer YfcM (Part 2/2)**
>
> >**Q3:** Similar as for the motivation, the results are lacking evidence that the proposed method alleviates the problem that is claimed to cause lower performance.
>
> **R:** We provide a performance analysis to support our claim more clearly, which is shown in Tab. 2. We first calculate the graph distances between each sample and all classes (detailed in App. 6.3) for CTR-GCN. Then, we rank the distances from low to high. In Tab. 2, we report the recognition accuracies of samples according to their distance ranks to the correct class. Here, higher ranks indicate that graphs are of higher quality (intra-class compact and inter-class dispersed), while lower ranks indicate that graphs are of lower quality (intra-class dispersed and inter-class aliasing).
>
> **Table 2: Performance (%) of samples with different graph distance ranks to the correct class.**
> |     Rank     | CTR   | CTR w/Ours | +/-  |
> |----------|-------|------------|-------|
> | rank1     | 97.2  | 97.4       | +0.2  |
> | rank2-5   | 92.1  | 92.9       | +0.8  |
> | rank6-10  | 88.3  | 89.3       | +1.0  |
> | rank11-20 | 84.9  | 86.3       | +1.4  |
> | rank21-40 | 83.2  | 84.8       | +1.6  |
> | rank41-60 | 80.8  | 82.9       | +2.1  |
>
> Several noteworthy observations are: (1) Considering samples from lower ranks to higher ranks, performances improve monotonically, revealing the significant correlations between graph quality and recognition performance. (2) Combined with the proposed method, we improve performances in all cases, where larger improvements are obtained on the samples with lower-quality graphs. These results prove that our method can alleviate the problem caused by learning low-quality graphs. We have included the results and analysis in our revision (Tab. 10 of Sec. 4.4).
>
> If the response satisfies your concerns, please consider reassessing the score. Thanks for your valuable comments and looking forward to your reply.

---

> ### Comment · Reviewer_YfcM · 2022-11-22
> **Response to Rebuttal**
>
> I would like to thank the authors for the careful rebuttal. After rebuttal, the authors are showing some initial evidence that support their claim a bit better. This clearly adds some value to the paper. The presented gains, both in the original paper and in the added experiments, are rather small though.
>
> I agree with reviewer Cq4q that small gains of ~1% as presented here are not convincing. I still think the paper is below the acceptance threshold.

---

> > ### Author Response · Authors · 2022-11-25
> > **Response to Reviewer YfcM**
> >
> > Thanks for your precious time in response to our rebuttal. We are delighted to find that our rebuttal has addressed your initial concerns about the novelty and evidence of the motivation.
> >
> > As for your newly raised concern about whether our ~1% improvement is significant and convincing, please allow us to explain this from two aspects:
> >
> > First, **~1% improvement in accuracy is significant and hard-won** in the field of skeleton-based action recognition. As a fundamental task in human action representation learning, skeleton-based action recognition has been extensively-benchmarked in recent years. In Table 1, we present the performance of several well-known methods published in top conferences for comparison. These methods were all state of the arts (SOTA) when publishing, where the improvements among them from 2019 to 2022 on NTU-based settings are between 0.2%~1.4%.
> >
> > **Table 1: Performance improvement (%) brought by previous SOTA methods.**
> > | Method    | Track    | NTU60-XSub-J            | NTU60-XView-J         | NTU120-XSub-J           | NTU120-XSet-J           |
> > |-----------|----------|-------------------------|-----------------------|-------------------------|-------------------------|
> > | 2S-AGCN   | CVPR2019 | 88.9                    | 94.5                  | 84.0                    | 85.3                    |
> > | MS-G3D    | CVPR2020 | 89.4(**+**$0.5$)   | 95.0(**+**$0.5$) | -                       | -            |
> > | MST-GCN   | AAAI2021 | 89.0(**+**$0.1$)   | 95.1(**+**$0.6$) | 82.8(**-**$1.2$) | 84.5(**-**$0.8$) |
> > | CTR-GCN   | ICCV2021 | 89.8(**+**$0.9$)   | 94.8(**+**$0.3$) | 84.9(**+**$0.9$)   | 86.7(**+**$1.4$)   |
> > | InfoGCN   | CVPR2022 | 89.4(**+**$0.5$)   | 95.2(**+**$0.7$) | 84.2(**+**$0.2$)   | 86.3(**+**$1.0$)   |
> >
> > Second, **the improvements are valuable** if in-depth investigations are involved. As presented in [Response to Reviewer YfcM (Part 2/2 Q3)](https://openreview.net/forum?id=PLUXnnxUdr4&noteId=Ptj3744TrNB), larger improvements are obtained on the **harder samples** with lower-quality graphs, which proves that our method can alleviate the problem caused by learning low-quality graphs (corresponding to our motivation). In addition, as presented in [Response to Reviewer Cq4q (Part 1/3 Q2)](https://openreview.net/forum?id=PLUXnnxUdr4&noteId=hZWt0O0gq7H) and [Response to Reviewer Cq4q (Part 2/3 Q3)](https://openreview.net/forum?id=PLUXnnxUdr4&noteId=hZWt0O0gq7H), the overall standard deviation in performance is quite stable and the more significant performance gains are achieved on **the hardest classes**.  Notably, our methods could be seamlessly integrated with graph-neural-network-based methods. Hence, such a design can provide technical insights for other graph-based tasks like molecular property prediction, protein function prediction, and traffic forecasting.
> >
> > Through comparison with well-known papers and in-depth performance analysis on hard samples, we attempt to make it clear that our improvements are solid and convincing. If the response satisfies your concern to some extent, please consider reassessing the score. Thanks for your valuable comments again.

---

> > ### Author Response · Authors · 2022-12-05
> > **Reliability of Performance Improvement**
> >
> > **R:** Thanks for reviewing our response and improving score. We sincerely appreciate your constructive suggestions during reviewing and discussion periods, which indeed helps us improve the quality of this paper, in terms of the explanation of motivation and interpretation of performance improvement.
> >
> > To further show the significance of our improvement, we have included experiments with different weight initialization for models.
> >
> > Taking the case of CTRGCN on the NTU60 dataset, we change the random seed to different values and run the experiments five times. Tab.1 shows the mean accuracy and standard deviation:
> >
> > **Table 1: Performance comparison with standard deviation.**
> > | Model | Accuracy (%) |
> > | -------- | -------- |
> > | CTRGCN     |   89.76 $\pm$ 0.07  |
> > | CTRGCN *w/SkeletonGCL* | 90.77 $\pm$ 0.06 |
> >
> > From the experiments, we can notice that the results are rather stable, which proves the reliability of the improvement. These results were also included in [Response to Reviewer Cq4q (Part 1/3 Q2)](https://openreview.net/forum?id=PLUXnnxUdr4&noteId=hZWt0O0gq7H).
> >
> > We hope the above response can alleviate your concern. Thanks again for your valuable comments.

---

### Official Review · Reviewer_YiMJ · 2022-10-24

**Confidence:** 4
**Correctness:** 3
**Technical Novelty And Significance:** 3
**Empirical Novelty And Significance:** 3
**Recommendation:** 8

**Clarity, Quality, Novelty And Reproducibility:**

**Clarity/Reproducibility:** as discussed in the strengths, the work is properly presented and discussed. All the most relevant points are defined with enough details, hence can be exploited to reproduce the submitted work (implementation settings are also properly listed).

**Quality:** the submitted paper covers a challenging task by proposing a simple, yet effective, approach that can be applied on top of the different existing solutions by introducing a negligible computational effort.

**Novelty:** in my personal view, the paper has properly clarified the differences with the existing literature thus making clear how it differs from the current works. Hence, the proposed approach is novel enough to possibly justify a publication.

**Strength And Weaknesses:**

**Strengths**
+ The paper is well-written, easy to follow, and presented concisely. All the most relevant details are described with enough depth providing the reader with enough information to properly grasp the overall idea and the rationale behind the choice of the implemented components. The implementation details seem to be enough to replicate the proposed solution. This would be very helpful to help further studies.
+ The approach seems quite simple, yet it shows interesting performance. The paper proposes to exploit inter-sequence information as well as intra-sequence features to learn a proper representation to recognize actions with a graph-based approach. This comes with the plus that, as shown in the experimental results, the approach adds negligible computational effort to current schemes.
+ The proposed approach can be applied on top of existing solutions. The method introduces a separate branch that exploits the graph learned through an existing approach (e.g., CTR-GCN, 2s-AGCN, etc.) to improve its generalization capabilities by exploiting the contrastive learning scheme.

**Weaknesses**

Overall, the paper does not have critical issues that may hamper a publication. There are a few minors that can be addressed with a review round. These include:
- While describing the evaluation methodology the paper states that there are four modalities being considered to compute the results, i.e., (J), (B), (J-B), (B-M), which can also be used together as in (4S). However, the tables (e.g., Table 1 and Table 2) report on the performance considering (J), (B), (J+B), and (4S). Clarifications/fixes on these are needed.
- There are a few hyperparameters listed in the experimental details that have no justification. These are the number of instances in each class $P$, the dimension of the graph vector $C_g$, and the dimension of the samplings for loss computation $K^H$'s and $K^R$ 's. It would be interesting to hear if these are found through experimental validation, cross-validation, etc.
- The submission does not come with a discussion about the limitations of the proposed approach. It would be interesting to hear what are the limiting factors that the authors came across.

**Summary Of The Paper:**

The paper introduces a graph neural network-based approach to tackle the skeleton-based action recognition problem. The work builds upon the intuition that current methods are not explicitly exploiting the information shared by all available sequences but are only using the intra-sequence features to learn proper graph representations. Following such an intuition, the paper formulates the problem of learning better representation through a contrastive learning approach that uses graph embeddings to pull graphs for the same action together, while pushing away graphs for other classes. This is combined in a solution that exploits a contrastive learning approach and a separated classification head. Experiments on 3 benchmark datasets show that the approach yields limited improvements with negligible computational cost.

**Summary Of The Review:**

I found the paper easy to read and well-motivated. The solution, despite being simple, provides (marginal) improvements to different existing approaches. This demonstrates the capabilities of the approach that can also be extended to other domains. The computational effort at training time is negligible and that there is no impact at inference time (wrt to the adopted base model). In light of all such considerations, the paper seems to have enough merits to justify a publication.

---

> ### Author Response · Authors · 2022-11-19
> **Response to Reviewer YiMJ**
>
> Thanks for your precious time reading this paper. We are delighted that you found the paper easy to read, well-motivated, capable to be extended to other domains and having enough merits for a publication.
>
> Please refer to the detailed response below, regarding the evaluation methodology, hyperparameter study, and discussion about limitations.
>
> >**Q1:** While describing the evaluation methodology the paper states that four modalities are being considered to compute the results, i.e., (J), (B), (J-M), (B-M), which can also be used together as in (4S). However, the tables (e.g., Table 1 and Table 2) report on the performance considering (J), (B), (J+B), and (4S).
>
> **R:** Thanks for your careful review. We have included additional statements in the revision to make it more clear as follows: in the field of skeleton-based action recognition, it is a widely-adopted protocol to evaluate models using *J*, *B*, *J+B*, and *4S* modalities.
>
> >**Q2:** These are the number of instances in each class $P$, the dimension of the graph vector $C_g$, and the dimension of the samplings for loss computation $K^H$'s and $K^R$'s. It would be interesting to hear if these are found through experimental validation, cross-validation, etc.
>
> **R:** We have included corresponding ablation experiments and analysis of hyperparameter settings in App. 6.2 of our revision.
> Tab. 1 shows the performance comparison using different values of $P$.
>
> **Table 1: Performance comparison with different $P$.**
> | $P$ | Acc (%) |
> | -------- | -------- |
> | 128 | 90.3 |
> | 342 | 90.5 |
> | **684** | **90.8** |
> | 1368 | 90.4 |
> | 2736 | 90.0 |
>
> Tab. 2 shows the performance comparison using different values of $C_g$.
>
> **Table 2: Performance comparison with different $C_g$.**
> | $C_g$ | Acc (%) |
> | -------- | -------- |
> | 64 | 90.4 |
> | 128 | 90.6 |
> | **256** | **90.8** |
> | 512 | 90.5 |
>
> Tab. 3 shows the performance comparison using different values of $K_H^+$.
>
> **Table 3: Performance comparison with different $K_H^+$.**
> | $K_H^+$ | Acc (%) |
> | -------- | -------- |
> | 64 | 90.6 |
> | **128** | **90.8** |
> | 256 | 90.6 |
> | 512 | 90.5 |
>
> Tab. 4 shows the performance comparison using different values of $K_H^-$.
>
> **Table 4: Performance comparison with different $K_H^-$.**
> | $K_H^-$ | Acc (%) |
> | -------- | -------- |
> | 128 | 90.5 |
> | 256 | 90.7 |
> | **512** | **90.8**  |
> | 1024 |  90.5 |
>
> Tab. 5 shows the performance comparison using different values of $K_R^-$.
>
> **Table 5: Performance comparison with different $K_R^-$.**
> | $K_R^-$ | Acc (%) |
> | -------- | -------- |
> | 128 | 90.6 |
> | 256 | 90.7 |
> | **512** | **90.8**  |
> | 1024 |  90.6 |
>
> >**Q3:** The submission does not come with a discussion about the limitations of the proposed approach. It would be interesting to hear what are the limiting factors that the authors came across.
>
> **R:** Thanks for your constructive suggestion. In this paper, we push away the negative pairs from different classes in the same way without considering their intrinsic relations. Take a sequence from class "walking" as an anchor. In such cases, sequences from class "running" and "swimming" are both negative samples. However, "running" is semantically much closer to "walking" than "swimming" is, therefore treating these negative samples coequally may neglect their intrinsic relations. Consequently, a comprehensive contrasting manner may be more suitable by delicately involving cross-class relations. We leave this for future work. Besides, we have included the limitation discussion in our revision (Sec. 5).
>
> We hope the explanations and results above solve your concerns. Thanks for your valuable comments and looking forward to your reply.

---

> > ### Comment · Reviewer_YiMJ · 2022-11-22
> > **Rebuttal**
> >
> > This reviewer acknowledges the efforts of the authors in revising the paper according to the raised concerns and suggestions (from all referees). I have sincerely appreciated the fact that experimental results have been run in a short amount of time with convincing results. The fact that authors have also considered some limitations of their work is a sign of great value for future readers. In light of such considerations, I am voting for acceptance of the submission.

---

> > > ### Author Response · Authors · 2022-11-25
> > > **Response to Reviewer YiMJ**
> > >
> > > Thanks for your recognition and kind words about our work. Your appreciation inspires us a lot.

---

### Author Response · Authors · 2022-11-19
**General Response: Paper Updates**

We sincerely thank all the reviewers and the area chair for their efforts in reviewing our paper. Below, we would like to highlight major paper updates during the discussion period.

* **Sec. 1:** Following the kind suggestion by Reviwer Cq4q, we have included the memory bank design as part of our contribution in the revision (end of Sec. 1).
* **Sec. 4.4**: Following the question raised by Reviwer YfcM, we have included quantitative results in the revision (in Tab. 9 of Sec. 4.4) to explain the strong relations between recognition performance and class-specific graph learning to support our hypothesis. Besides, we also include performance analysis in the revision (in Tab. 10 of Sec. 4.4), to make it more clear how our method alleviates the problem we propose (Reviewer YfcM) and improves the performance on hard examples (Reviewer Cq4q).
* **Sec. 5:** Following the kind suggestion by Reviwer YiMJ, we have included a limitation discussion in the revision (Sec. 5).
* **App. 6.2**: Following the kind suggestion by Reviewer YiMJ, we have included several ablation experiments on the hyperparameter settings in the revision (Tab. 15, 18, 19, and 20 of App. 6.2). In addition, we provide performance analysis on hard examples in the revision (Tab. 21 and Tab. 22 of App. 6.2) for Reviewer Cq4q.

* **App. 6.3:** We elaborate on how we measure graph distances in the revision (App. 6.3) for Reviewer YfcM and Cq4q.

Thanks,

Paper578 Authors

---

### Author Response · Authors · 2022-12-12
**To All Reviewers and Area Chairs**

Dear reviewers and area chairs,

We sincerely thank all reviewers and area chairs for their valuable time and comments. During the two-stage discussion, we responded to all reviewers’ comments and provided experiments/analyses/clarifications to solve their concerns. Here, we summarize why ~1% improvements are significant in terms of the following four aspects:

* By comparing with the improvements of previous state-of-the-art methods, it shows that **~1% improvements in accuracy are hard to achieve in the field of skeleton-based action recognition**. Please refer to [Response to Reviewer YfcM](https://openreview.net/forum?id=PLUXnnxUdr4&noteId=YqTGZlF6j7) for more details.
* We run experiments several times with different weight initializations for models. From the experiments, it is noticed that **the standard deviation of results are rather low, which means the performance is stable and proves the reliability of the improvements**. Please refer to [Reliability of Performance Improvement](https://openreview.net/forum?id=PLUXnnxUdr4&noteId=tGWjCoCv4P) for more details.
* We show that our method can **significantly improve the hardest classes and samples**. These results further demonstrate the effectiveness of our method and support our hypothesis and motivation. Please refer to [Response to Reviewer Cq4q (Part 2/3)](https://openreview.net/forum?id=PLUXnnxUdr4&noteId=Id1EFa4SG88) for more details.
* It is notable that the lifting on performance with our method is **consistent for all three popular GCN-based methods**. Considering NTU60, NTU120, and NW-UCLA datasets are extensively benchmarked, such consistent improvements are hard-won and become more significant. Please refer to Tab.1 and Tab.2 of the paper.

Thanks again for the efforts of all reviewers and area chairs.

Best,

Paper578 Authors

---

### Decision · Program_Chairs · 2023-01-20

**Decision:**

Accept: poster

**Justification For Why Not Higher Score:**

Though the method is interesting and novel, the main weakness is that performance gain is not very obvious.

**Justification For Why Not Lower Score:**

It is an interesting perspective to take advantage of the cross-sequence context information to enhance the graph discrimination capability.

**Metareview: Summary, Strengths And Weaknesses:**

This paper proposes to better leverage the global context (intra-sequence context) to construct adaptive graphs for skeleton based human action recognition. The designed method associates graph learning over different sequences and drives the graphs to be class discriminative, which improves the graph convNet's capability to distinguish various action patterns. Most of the reviewers (and also the area chair) agree that this view is interesting. Specifically, the performance improvement on hard classes is good. However, the major weakness is that the overall performance gain on the whole dataset is ~1% only. Yet, considering that the performance improvements reported by state-of-the-arts are also at a similar scale, and also, the method obtains more obvious improvement on hard classes, the AC recommends accept for this paper. Authors need to incorporate the new results and content provided in rebuttal to the final version of the main paper.

**Note From Pc:**

if the above contains the word "oral" or "spotlight" please see: "oral" presentation means -> notable-top-5% and "spotlight" means -> notable-top-25%. As stated in our emails, we are disassociating presentation type from AC recommendations